# Understanding the Evolutionary Ecology of host–pathogen Interactions Provides Insights into the Outcomes of Insect Pest Biocontrol

**DOI:** 10.3390/v12020141

**Published:** 2020-01-25

**Authors:** David J. Páez, Arietta E. Fleming-Davies

**Affiliations:** 1School of Aquatic and Fishery Sciences, The University of Washington, 1122 NE Boat St, Box 355020, Seattle, WA 98195, USA; 2Department of Biology, University of San Diego, 5998 Alcala Park, San Diego, CA 92110, USA

**Keywords:** transmission, heterogeneity, tradeoffs, fecundity, genetic variation, insect pathogens, population dynamics

## Abstract

The use of viral pathogens to control the population size of pest insects has produced both successful and unsuccessful outcomes. Here, we investigate whether those biocontrol successes and failures can be explained by key ecological and evolutionary processes between hosts and pathogens. Specifically, we examine how heterogeneity in pathogen transmission, ecological and evolutionary tradeoffs, and pathogen diversity affect insect population density and thus successful control. We first review the existing literature and then use numerical simulations of mathematical models to further explore these processes. Our results show that the control of insect densities using viruses depends strongly on the heterogeneity of virus transmission among insects. Overall, increased heterogeneity of transmission reduces the effect of viruses on insect densities and increases the long-term stability of insect populations. Lower equilibrium insect densities occur when transmission is heritable and when there is a tradeoff between mean transmission and insect fecundity compared to when the heterogeneity of transmission arises from non-genetic sources. Thus, the heterogeneity of transmission is a key parameter that regulates the long-term population dynamics of insects and their pathogens. We also show that both heterogeneity of transmission and life-history tradeoffs modulate characteristics of population dynamics such as the frequency and intensity of “boom–bust" population cycles. Furthermore, we show that because of life-history tradeoffs affecting the transmission rate, the use of multiple pathogen strains is more effective than the use of a single strain to control insect densities only when the pathogen strains differ considerably in their transmission characteristics. By quantifying the effects of ecology and evolution on population densities, we are able to offer recommendations to assess the long-term effects of classical biocontrol.

## 1. Introduction

Classical biocontrol refers to the control of pest species by the deliberate introduction of exotic or adapted natural enemies including pathogens, parasites, herbivores or predators [1]. However, just under a third of classical biocontrol attempts targeted at insect pests have been successful [2]. Many of these failures have been attributed to the low rate of establishment of introduced control agents [3,4,5], but cases where the control agent became established and yet failed to depress the targeted insect pest population are also common. These failures result from a multitude of factors, ranging from management decisions [3], to host shifts of control agents in novel ecological communities [6], to unexpected consequences of pest and enemy life-history plasticity and evolution [7,8].

To maximize the success of control programs, ecologists and other stakeholders evaluate both the risks of introducing control agents in new ecological settings and the likelihood that the control agent will successfully diminish or completely suppress the targeted pest species’ population. Such testing procedures are grounded in evolutionary and ecological principles, as they require the analysis of life-history traits in the pest and control agent and of potential ecological interactions with other non-targeted species under different environmental conditions [9]. To assess the potential for long-term success of proposed control programs, this ecological knowledge is then used to develop demographic predictions of the pest species by modeling the population dynamics of the insect pest and control agent [10].

Most of these modeling efforts exclude evolutionary effects, as it is often assumed that evolutionary change occurs over much longer time-scales than ecological change. However, several lines of evidence suggest that insect pests and their control agents can coevolve over the ecological time considered in control programs [11,12] and thus affect the outcome of biocontrol [8,13,14]. First, there is abundant evidence of substantial genetic variation in relevant traits in both the insect host and potential control agent [15,16,17,18,19,20,21]. In addition, generation times are typically quite short in pest insects, and empirical studies have documented rapid evolution in various invertebrate host species in response to natural enemies, including *Drosophila*-parasitoid interactions [22], *Daphnia*-parasite interactions [23], and social insect immune systems [24]. Although there is an increasing interest in understanding how ecology and evolution interact to mediate population dynamics of invertebrate hosts and their natural enemies [15,23,25], few efforts have been made to incorporate relevant evolutionary mechanisms to develop long-term predictions applicable to biological control (but see [26,27,28] ), making it difficult to evaluate whether such mechanisms should be considered during the development of control programs.

Pathogens, including viruses, are important natural enemies of many insect pests [29,30]. Viruses are attractive as biocontrol agents because they are easily deployed, can sustain prolonged suppression of insect populations without repeated interventions, and are often species specific, thereby minimizing negative effects on non-targeted species [29,31,32,33]. The models that we consider are based on the host–pathogen interactions between insects and baculoviruses, DNA viruses that primarily infect Lepidoptera [33]. Horizontal transmission of baculoviruses occurs when larvae consume infected food substrates. Infected larvae die after an incubation period and release viral particles onto those same substrates, where other healthy larvae may encounter the virus and become infected. If the epidemic does not ‘burn out’ from the lack of susceptible hosts, it is ended when hosts pupate because many pest Lepidoptera lose the ability to feed during adult stages or become more resistant to infection with increasing age [33,34]. Our models also assume that the insect–virus interactions occur in the temperate regions and thus insect hosts experience on average one generation per year [35]. We thus assume that infection is not possible during insect diapause over winter when hosts do not consume food. Although there is evidence of vertical transmission in baculoviruses [33], we assume that most transmission across seasons is due to overwintering by the virus on surfaces outside of the host, e.g., [36]. At the end of the season, surviving insects reproduce, determining the number of larvae produced in the next generation. The following spring, healthy larvae hatch from eggs and consume virus that has persisted from the previous generation. Although the models considered here can be modified to include regulatory effects of other organisms, such as defense mechanisms in plants [37] and other parasites or predators [15,38], here we assume that the insect population is exclusively regulated by the viral pathogen.

Several features of insect pests and the pathogens used in biocontrol suggest that evolutionary processes may play an important role in their interaction. Large population sizes and rapid generation times of pest insects and their viral pathogens help maintain the genetic variation that is the target of natural selection. Pathogens used in insect biocontrol are often lethal, suggesting a strong selective pressure for insect survival. For seasonal host–pathogen interactions, selection may also operate on pathogen survival outside the host over periods of insect diapause, in addition to maximizing transmission during periods of insect activity. Genetic drift is also likely to affect insect and pathogen populations. In insect hosts, drift is likely to occur after a collapse in the insect population. Viruses experience bottleneck and drift effects during host colonization that likely affect the outcome of within-host processes determining infection success [28,39].

In this study, we use mathematical models specific to insect–virus interactions to explore the effects of eco-evolutionary interactions on population control of insect pests. We focus on classical biocontrol scenarios, in which interactions between the control agent and target pest more closely mimic natural host–pathogen population dynamics, while excluding other forms of biocontrol such as pheromones and bio-toxins [1]. We consider two key eco-evolutionary processes: host and pathogen heterogeneity and life-history tradeoffs. For each of these topics, we review the existing literature, summarize prior mathematical models incorporating these evolutionary processes, and then use numerical simulations of population models to explore their consequences for biocontrol. We end with recommendations for using eco-evolutionary process models in classical biocontrol.

## 2. Heterogeneity in Pathogen Transmission

It is well known that there is variation among hosts in the likelihood of contracting infection and of transmitting infection to other hosts [40,41,42,43,44]. Heterogeneity in transmission can stem from variation in spatial processes affecting host contact rates i.e., movement patterns, dispersal, and the availability of food and reproductive habitats; [45,46,47] and the structure of contact networks connecting infected and susceptible hosts [48,49]. Heterogeneity in transmission can also result from differences between individuals in either susceptibility or infectiousness, as a result of variation in immune system functioning [50,51], age or developmental stage effects [20,52], pathogen avoidance [53,54], or other genetic and environmental effects. While both spatial effects and individual variation are important contributors to heterogeneity in transmission in insect–pathogen systems, our study will focus on the effects of individual variation arising from genetic, environmental or stochastic effects.

Insects display widespread variation in pathogen defense traits, including disease clearing [55,56] and behavioral mechanisms of disease avoidance [53,57]. In the simplest case, variation in disease outcomes among individuals can stem from stochastic processes within the host, as pathogen populations in a single host grow or decline from a small initial founding population upon exposure [28,39]. Previous studies have also shown that at least part of the variation among individual insects is genetically based [15,16,58]. In addition to genetic effects, insect immune systems can vary due to trans-generational epigenetic effects (sometimes called “priming”) [59], in which parental exposure to sub-lethal doses of pathogen produces lower infection probability in offspring [60,61]. However, there is limited evidence demonstrating that priming is widespread across insect species [16,62] or that insect population dynamics are regulated by these effects (but see [63,64] for models).

When heterogeneity in transmission is caused in part by genetic differences between individuals, this allows for the average transmission rate in the population to change over time due to evolutionary forces such as natural selection or genetic drift. Recent work has indeed shown significant heritability of transmission in an insect baculovirus [15], lending support to the hypothesis that natural selection could favor lower transmission under high pathogen density. This work also suggested that the continual evolution of lower transmission could be constrained by costs of increased immune defenses on fecundity, so that under low pathogen densities, natural selection would favor higher transmitting, more fecund individuals (see section *Ecological and Evolutionary Tradeoffs* below for more details). Natural selection on transmission rates is further likely to drive the population dynamics of insect host–pathogen systems for the following reasons: First, insect pathogens used in biocontrol, such as baculoviruses, are often lethal, leading to strong selection on hosts for survival and thus lower susceptibility to infection. Second, many of these pathogens are only transmitted upon host death, causing similarly strong selection on pathogens for high transmission rates, and increasing the likelihood of a coevolutionary ‘arms race’ between host and pathogen [65,66]. Third, in insect–virus systems, both host and pathogen typically have short generation times, suggesting that rapid rates of evolutionary change are likely, with changes occurring at similar time scales to population dynamics.

### 2.1. Modeling Heterogeneity in Transmission in Insect–Pathogen Systems

As shown in the previous section, variation in host–pathogen interactions has been quantified in a variety of systems and traits. However, to understand how variation affects pest insect population sizes, we need to characterize this heterogeneity in terms of disease transmission which can then be incorporated into population dynamics models. Most mathematical models of insect–pathogen population dynamics assume density dependent transmission, modelling this transmission as a mass-action process that is directly proportional to the densities of susceptible and infected hosts [67,68]. In the foundational Susceptible-Infected-Recovered (SIR) ordinary differential equation models, density dependent transmission is simply βSI, where β is the transmission rate and *S* and *I* are the densities of susceptible (or healthy) and infected hosts, respectively [69]. However, empirical studies often demonstrate that the infection process is non-linear with respect to host densities [70,71], for example, with changes in pathogen densities causing only small changes in the number of infected hosts.

A crucial result obtained by fitting SIR-type models to data on infectious diseases is the discovery that the non-linear relationships between host density and transmission are often better explained by models that incorporate heterogeneity in transmission [44,71,72,73,74]. Heterogeneity in transmission has been incorporated in these models by including more than one class of hosts (e.g., by assuming that susceptible or infectious individuals are classed in different groups based on age, behaviour or other phenotypic classes [75,76]), or by assuming that the transmission rate is distributed continuously and incorporating a model parameter representing the variation of this distribution (e.g., in humans [43]; fish [77]; and insects [44]).

In the models that we present here, we adopt the latter approach, assuming that host susceptibilities are Gamma distributed so that the distribution of transmission rates is approximated by the mean transmission $\bar{\nu }$ and the coefficient of variation of transmission *C* ([44,72,78], see Box 1 for details). Thus, these models do not require an explicit mechanism for the processes generating heterogeneity, which might include stochastic, environmental, and genetic sources.

One key advantage of this structure is that it is possible to make general predictions for how host variation might impact insect population dynamics in virus biocontrol programs even in the absence of specific knowledge on how individual variation is generated in a particular insect–pathogen system. In addition to this general structure to incorporate heterogeneity effects, we consider effects of specific processes generating heterogeneity, including genetic variation in host susceptibility to one or multiple pathogen strains. We also discuss how variation in susceptibility generates tradeoffs that affect insect population densities and host–pathogen population dynamics (Table 1).

Box 1Heterogeneity in transmissionThe models presented here incorporate heterogeneity as a continuous Gamma distribution described by the mean transmission ν¯ and the squared coefficient of variation of transmission C2 (which is unitless). We term this heterogeneity in transmission rather than host heterogeneity to recognize that it is a trait of both pathogen and host and can arise from various sources, including within-host stochastic processes [28], host behavior [53], environmental factors [80], as well as both host and pathogen genetics and G × G interactions [15,16,17,63,79]. When heterogeneity C2 equals 0, all individuals are identical in their susceptibility to disease and the infection process will be completely driven by the mean transmission rate. When heterogeneity is greater than 0 (C>0), then there are differences between individuals in susceptibility, with some individuals being more susceptible to infection than others (Figure 1).Figure 1Simplified schematic of host heterogeneity, showing just two types of hosts, high susceptibility (red) and low susceptibility (blue). As the disease spreads, the high susceptibility hosts are infected first, lowering the average susceptibility (and thus the mean transmission ν¯(t)) of the population.
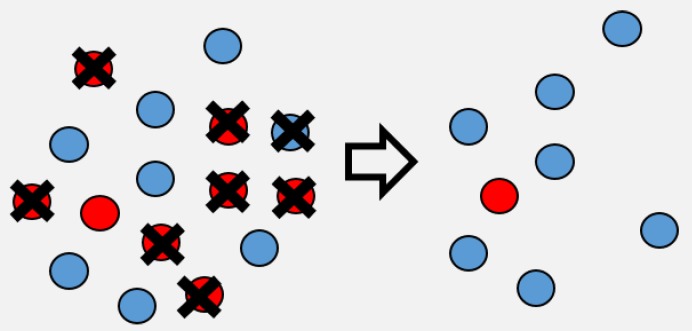
Assuming a continuous distribution of host susceptibilities, as the epidemic progresses, the most susceptible hosts are removed first, and thus the individuals left in the population are those of lower than average susceptibility (Figure 1). The transmission rate thus drops as it becomes more and more difficult for the pathogen to infect the remaining hosts (Figure 2). Comparing two host populations with different levels of heterogeneity, the range of host susceptibilities is narrower in the population with a lower heterogeneity of transmission C2. Thus, the individuals that were removed are more similar to the individuals remaining uninfected, and the transmission rate ν¯(t) does not drop as dramatically. This process is essentially strong selection for low host susceptibility and has a strong effect on host density (Figure 3). In a model incorporating host evolution (Table 1, Model 3), the distribution of insect susceptibilities is characterized by genetic variation and so transmission is allowed to evolve from host generation to host generation. Evolution of ν¯ towards infinitely small values is however constrained by tradeoffs between transmission and insect fecundity (see Box 2). By contrast to the evolution model, in the other models (Table 1, Models 1-2,4) the mean transmission value starts at the same value of ν¯ at the beginning of the next epizootic. Thus the evolution model adds realism by allowing the hosts to evolve.Note that heterogeneity is also a trait of the infecting pathogen. Two pathogen strains might result in two different distributions of susceptibilities in the same host population, if the hosts are more variable in their susceptibilities to one strain than the other. Thus, the two examples in Figure 2 could also be produced by two different pathogen strains in the same host population. In the two-strain model, heterogeneity C2 is attributed to pathogen strain as the strains share a single host population (Table 1, Model 4). Host evolution is excluded from this model for simplicity, and the mean transmission rate ν¯ is assumed to reset at the same value at the beginning of each outbreak.Figure 2As the epidemic proceeds, the instantaneous mean transmission ν(t) drops off as the more susceptible individuals are removed. This decrease is of larger magnitude for a more variable host population (red) than a less variable one (blue).
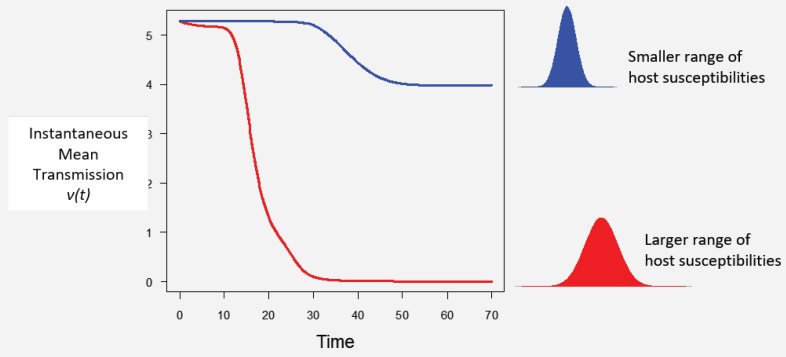
Figure 3Numerical simulation of a single epidemic of the one strain heterogeneity model (Model 2 in Table 1) for different values of heterogeneity of transmission C2. Colored lines show the population density of healthy insects and grey lines show the population density of pathogen over a period characteristic of the duration of larval stages of some Lepidoptera.
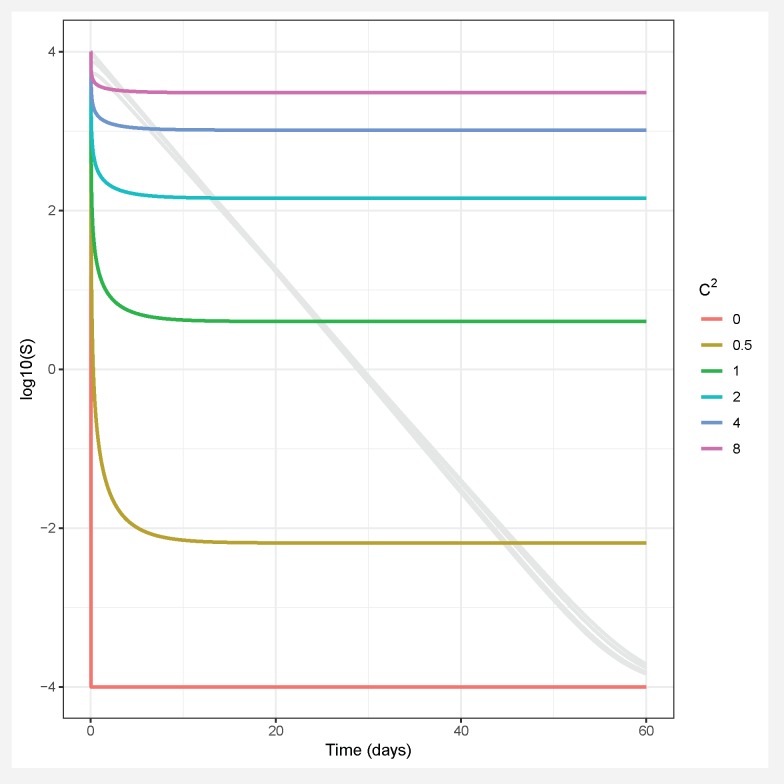


### 2.2. Consequences of Heterogeneity in Transmission for Biocontrol

Heterogeneity in transmission can have drastic effects on pest-insect biocontrol programs by directly influencing the number of hosts killed by pathogens in a single epidemic (Box 1, Figure 3) or over multiple generations. Prior empirical work has found that heterogeneity in transmission tends to correlate with the mean transmission value (as shown in Box 2) [79]. This correlation forms the backbone of a life-history tradeoff discussed in the following section. However, it is possible to evaluate the effects of heterogeneity on insect population size independently from this tradeoff, by fixing the average transmission rate to a particular value and varying the heterogeneity of transmission. In the absence of tradeoffs, higher heterogeneity of transmission is associated with higher mean long-term insect population density for any given mean transmission value (Figure 4a).

In fact, the magnitude of the effect of heterogeneity on long term insect population density is much larger than the effect of changing mean transmission rate over a similar ecologically reasonable range of parameter values (Figure 4a). This suggests that the natural variation that allows some insects to escape infection is more important in determining insect population sizes than the average number of insects that die from infection.

In addition to affecting average insect population density, increased heterogeneity also affects the stability of long-term population dynamics. For example, higher heterogeneity shifts population densities from “boom–bust" cycles to constant insect populations over time in all one-strain models (Figure 5, Models 1–3, [72]). Furthermore, in cases where population cycles occur, increased heterogeneity increases the frequency but decreases the severity of insect outbreaks (as indicated by the effects of heterogeneity on the period and amplitudes of population cycles for one-strain models; Figure 5 and Figure 6). This is because higher heterogeneity allows more insects to escape infection, and thus the population takes less time to reach the densities required for another epidemic to occur. This effect is heightened in models where transmission evolves due to natural selection, as evolution of decreased susceptibility allows even more individuals to escape infection with each passing generation (Figure 6). However, it is important to consider that the models simulated here are all deterministic ODE models. In more realistic models allowing for stochastic processes, the large-amplitude cycles at low heterogeneity values could lead to extinction of either pathogens or hosts when populations crash to low levels. For example, stochastic models of human diseases that incorporate heterogeneity of infectiousness found that higher heterogeneity was associated with an increased probability of pathogen extinction, in spite of also leading to higher peaks of infection [43]. Thus, lower amplitude cycles at higher heterogeneity values might not be a positive outcome for biocontrol if they prevent population crashes that result in the extinction of small insect populations due to stochastic effects [81].

Heterogeneity in transmission thus has fundamental consequences for the effectiveness of biocontrol. While higher heterogeneity is associated with higher average insect population densities, the effects of decreased cycling under high heterogeneity in single pathogen strain systems would prevent the population density spikes that are commonly associated with outbreaking insects. Below, we will discuss how these predictions change with the added complications of ecological and evolutionary tradeoffs.

## 3. Ecological and Evolutionary Tradeoffs

Across many taxa, research has revealed how hosts compromise key life history processes such as homeostasis, growth and reproduction to combat pathogens and overcome infection [82,83]. Hosts can suffer costs through harmful effects resulting from immune system activity [84] and through reduced competitive ability due to diminished resource allocation to other life-history process while preventing infection [22,82,83,85,86]. Indeed, host behaviors that minimize pathogen exposure, such as pathogen avoidance [53,57], could result in reduced access to high quality food and reproductive habitats, or even reduced immunocompetence [54]. The magnitude of these costs to fitness depends on environmental context [87,88,89], but also varies greatly among individuals, likely generating heterogeneity in transmission. For example, a chief physiological process to combat pathogen infection in insects involves activation of the phenoloxidase pathway. However, the byproducts of phenoloxidase activity are toxic [90], causing harm to the host’s cells that are likely costly to overcome [84] and affecting reproductive investment [91]. Variation in phenoloxidase activity among individuals [18] thus not only generates variation in transmission in insects, but could also result in a tradeoff between transmission and fecundity [15].

Similarly, pathogens display tradeoffs between their own key life-history traits, such as between virulence (i.e., the damage or harm to the host resulting from infection) and transmission [92], or between survival outside the host and transmission [79]. Studies have also revealed how costs in overcoming host defenses guide the ecology of pathogens. For example, in the fungal plant pathogen *Melampsora lini*, virulent pathogen genotypes often dominate in host populations that have high resistance. However, overcoming host defenses for this pathogen comes at a cost of producing fewer spores, so that host populations characterized by high susceptibility are dominated by avirulent, more-fecund pathogen genotypes [93].

### 3.1. Modeling Tradeoffs with Transmission in insect–pathogen Systems

The evolution of virulence is often hypothesized to be constrained by transmission, forming the basis of a hypothesis known as the transmission-virulence tradeoff that has been the focus of many foundational mathematical models [94,95]. While this hypothesis has been used to understand coevolutionary dynamics [96,97,98] and the effects of virulence evolution on population dynamics [98,99,100,101], transmission is not always correlated with virulence [102,103] and few studies have determined how this tradeoff affects insect pest biocontrol. More generally, a difficulty in evaluating the role that tradeoffs play in the control of insect populations with pathogens is that tradeoffs are often evaluated with proxy measures of pathogen fitness and the exact relationship between these proximate measures and transmission are unknown. A strategy thus taken by some studies is to design experiments guided by the models used to understand disease spread and long-term population dynamics [72,104,105] to directly quantify relationships between transmission parameters [15,63,79,92,106,107].

Thus, to highlight applications to biocontrol, we focus on how tradeoffs in both hosts and pathogens have been incorporated in ecological models to understand population dynamics of insect-baculovirus interactions. Specifically, we consider tradeoffs between insect fecundity and transmission and between average transmission and heterogeneity in transmission, as defined above (see also Box 2).

The tradeoff between fecundity and transmission was quantified with prior field experiments on numerous full-sib families of the gypsy moth [15]. Specifically, low susceptibility to the gypsy moth’s lethal baculovirus (i.e., high resistance) was found to be correlated with decreased fecundity in a linear fashion, suggesting a fitness cost to low susceptibility. This raised the possibility that while increased pathogen density would select for lower transmission values (given that transmission is heritable), fecundity costs would prevent the evolution of transmission towards infinitely small values. These processes were then incorporated into a mechanistic model (Model 3, Table 1) to evaluate the potential consequences on long-term population dynamics. The resulting model produced a mechanism by which the average transmission changes from generation to generation (Figure 8).

In this model, no change in transmission occurs under weak selection and small fecundity costs (white curved line in Figure 8); transmission increases in the next generation when there is weak selection and high fecundity costs (pink-orange area in Figure 8); transmission decreases in the next generation with strong selection and small fecundity costs (dark purple in Figure 8); and transmission also decreases in the next generation under strong selection and large fecundity costs, but to a lower extent than under strong selection and small fecundity costs (lighter purple in Figure 8). An important shortcoming of this evolution model, however, is that heterogeneity, whether genetic or environmental, remains constant over time [15]. Although some theoretical work shows that reductions in population size or strong selection can erode genetic variation (but see, [108]), as far as we know, models in disease ecology have not addressed this issue.

The second tradeoff that we considered here was between average transmission $\bar{\nu }$ and heterogeneity in transmission $C^{2}$ (Box 2). This results in a life history tradeoff for the pathogen because (as discussed in the previous section; see Box 1), increased heterogeneity causes a more rapid decline of the transmission rate during the epidemic as the most susceptible individuals are removed from the population. A strain with high mean transmission but high heterogeneity will dominate early in the epidemic, but as the epidemic progresses, more insects will escape infection due to the high heterogeneity. In contrast, a low mean transmission, low heterogeneity strain will infect fewer individuals early in the epidemic, but will continue to do so more consistently as the epidemic progresses. Thus, a low mean transmission, low heterogeneity strain has similar fitness to a high mean transmission, high heterogeneity strain. Prior empirical work found evidence of this tradeoff between transmission and heterogeneity of transmission in gypsy moth baculovirus isolates, and this relationship was crucial for the coexistence of multiple pathogens in a single population [79].

Of the four insect–pathogen population models we compared, three incorporated a tradeoff between average transmission $\bar{\nu }$ and heterogeneity of transmission $C^{2}$ (Table 1, Models 2-4). In this pathogen life-history tradeoff, a strain with high average transmission $\bar{\nu }$ is associated with higher heterogeneity of transmission $C^{2}$. In two of these (Models 2 and 4), this tradeoff was incorporated explicitly, using a tradeoff curve fit to empirical data from multiple isolates of gypsy moth baculovirus [79]. In the host evolution model (Model 3), a similar positive relationship between heterogeneity of transmission and mean transmission occurred as host susceptibility evolved over generations, affecting both of these parameters. Thus the host evolution model incorporated two tradeoffs: between host fecundity and transmission [15], and between heterogeneity of transmission and mean transmission.

Box 2Transmission-heterogeneity tradeoffLife history tradeoffs occur when an evolutionarily advantageous trait comes at the cost of lower fitness in another trait. In a plot of two traits that both influence fitness, the presence of a correlation between traits may indicate presence of a tradeoff constraining fitness. This correlation may be positive or negative depending on whether higher or lower trait values are associated with higher fitness (Figure 9). For example, in the classic evolution of virulence tradeoff [94,95], both host mortality and between-host transmission tend to increase with higher rates of pathogen replication within the host (but see, [102,103]). Higher transmission is associated with higher pathogen fitness, but higher host mortality is associated with lower pathogen fitness. Thus, the presence of a positive correlation in a plot of pathogen transmission against host mortality (virulence) is consistent with a tradeoff. In the absence of tradeoffs, optimal pathogen fitness would occur at the highest possible values for transmission but the lowest values of mortality. However, the positive correlation between these two traits constrains their evolution such that high-transmission (higher fitness) strains also induce high host mortality (lower fitness). Thus, this correlation indicates the presence of a tradeoff.Figure 9Presence of a linear correlation between life history traits can indicate a tradeoff constraining the evolution of those traits. In the absence of tradeoffs, the highest fitness point in the parameter space presented above is indicated by the star. The slope of the line can be negative (**left panel**) or positive (**right panel**), depending on whether higher or lower values of each trait are associated with higher fitness, but critically, it does not cross the area of highest fitness for both parameters (star).
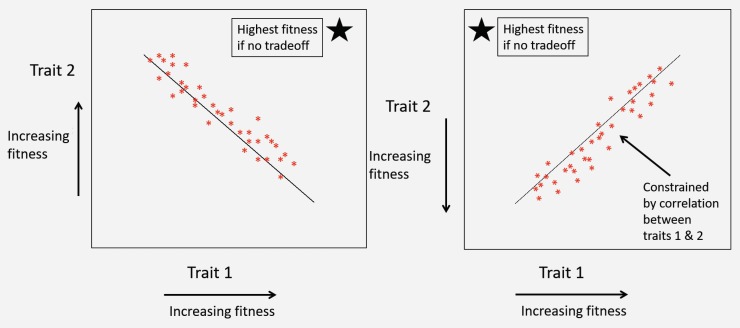
The key pathogen life history tradeoff we discuss in this paper is one we term the transmission-heterogeneity tradeoff. Heterogeneity of transmission C2 is described by the coefficient of variation (seeBox 1), and its positive correlation with the mean transmission rate ν¯ indicates a pathogen life-history tradeoff (Figure 10). This tradeoff has been observed empirically in the baculovirus that infects gypsy moths, using different field-collected strains [79], and likely arises in part due to interactions with host genotype; high transmission rate in some hosts comes at a cost of reduced ability to infect others [17] (see also Box 1). Intriguingly, a similar tradeoff between mean transmission and heterogeneity arises via a different mechanism in the host evolution model (Model 3 in Table 1), as an effect of the host life-history tradeoff between fecundity and disease resistance. This is because higher heterogeneity in that model is in part attributed to host genetic variation, so that at high heterogeneity, hosts evolve more rapidly to high transmission values, because the associated high fecundity is so advantageous [15].Figure 10Tradeoff between mean transmission rate ν¯ and heterogeneity of transmission C2. Higher values of heterogeneity C2 (the squared coefficent of variation of transmission, and thus unitless) leads to lower infection of hosts (see Box 1). The specific function shown here was fit to data from 16 strains of the NPV that infects gypsy moths, *Lymantria dispar* [79], but the tradeoff itself and general shape of the curve appear to be consistent features of the models used to describe host heterogeneity (Table 1).
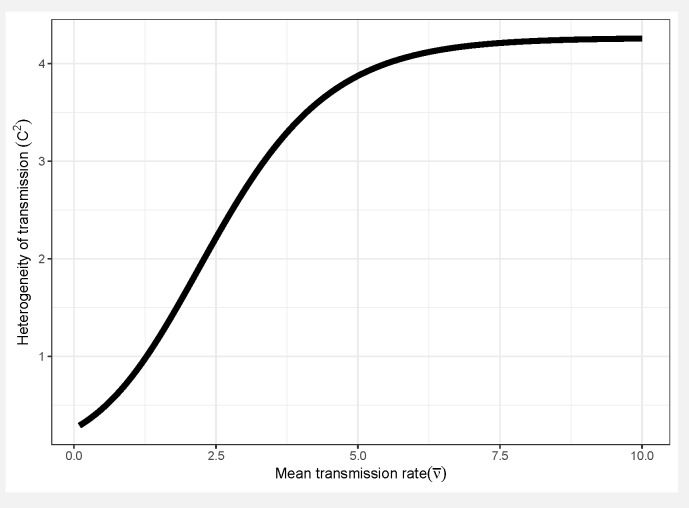


### 3.2. Consequences of Tradeoffs for Biocontrol

The transmission-heterogeneity tradeoff has strong effects on biocontrol effectiveness because mean transmission and heterogeneity in transmission have opposite effects on the long-term insect population density (Figure 4a,b). The stronger this tradeoff becomes, the weaker the effects of heterogeneity $C^{2}$ on insect density. Excluding the tradeoff effect (i.e., by fixing the average transmission at $\bar{\nu }=0.7$ while independently varying heterogeneity $C^{2}$; Model 1) shows that increasing heterogeneity in transmission results in higher long-term insect population densities (Figure 1b, red line). As discussed above, this is because higher variation in transmission in the absence of tradeoffs allows more insects to escape infection. Adding the transmission-heterogeneity tradeoff shows a similar overall increase in insect density with increasing heterogeneity of transmission, but this increase in density occurs more slowly (i.e., slope of green line, Model 2, versus red line, Model 1, Figure 4b). This is because in the presence of the tradeoff, larger heterogeneity values are associated with higher mean virus transmission, which suppresses insect populations. The non-linear shape of the curve (Figure 4b, green line) arises because the tradeoff has a sigmoidal shape (Box 2, Figure 10, [79]), so that the mean transmission $\bar{\nu }$ changes rapidly at low values and high values of heterogeneity $C^{2}$, and less rapidly in the intermediate range.

The host evolution model (Table 1, Model 3) also incorporates a tradeoff between mean transmission and heterogeneity in transmission, but the specific tradeoff function is different from that used in the other models (Table 1, Models 1, 2, 4) because it arises from the life history tradeoff between mean insect fecundity and mean transmission (e.g., higher host resistance comes at a cost of lower fecundity [15]). Heterogeneity in this model affects not only the spread of disease in the population, but also the capacity for evolutionary change in the host, because heterogeneity is attributed in part to genetic variation in the host population [15] (Box 2). Thus in the high heterogeneity case, hosts quickly evolve high mean transmission (i.e., high susceptibility) at low pathogen densities, because high susceptibility is associated with higher reproductive output, and genetic heterogeneity allows for evolutionary change. Despite the higher fecundity with increasing heterogeneity of transmission, the mean insect density decreases because mean transmission rate is increasing so rapidly due to the strength of the transmission-heterogeneity tradeoff in this model (Figure 4b, teal line).

In the two pathogen model (Table 1, Model 4), the effect of the tradeoff was evaluated by fixing the average transmission and heterogeneity in transmission of one strain to specific values ($\bar{\nu }=0.7$, $C^{2}=0.576$), and varying these parameters for the second strain according to the transmission-heterogeneity tradeoff function used in Model 2. In this model, insect density displays decreasing and then increasing phases as the heterogeneity of transmission of the second strain increases (Figure 4b, purple line). Below a heterogeneity of approximately $C^{2}=1.5$ in the second strain, two strain coexistence does not occur for these parameter values. As the heterogeneity of transmission of the second strain increases, the two strains become more phenotypically distinct, and thus two-strain coexistence is increasingly favored [79]. Coexistence is possible even for very low values of heterogeneity $C^{2}$ in one strain, as long as the other strain’s heterogeneity is substantially larger [79]. The effect of adding an additional strain that is able to infect a different range of hosts than the first is to suppress insect density, and thus the initial decrease in insect density with higher heterogeneity is observed. However, above approximately $C^{2}=2.6$, the direct effect of increased heterogeneity in transmission (as observed in the single strain Models, 1 and 2), overrides the effect of added transmission by the second strain, resulting in higher insect densities (Figure 4b).

Due to the common “boom–bust” population cycles of outbreaking forest insect populations, it is important to consider not only effects on the average insect population density, but also effects on the stability of insect population densities (i.e., cycling or constant over time) (Figure 5 and Figure 6). Insect population cycles occur over a substantial range of heterogeneity values for all of the population models considered here (Figure 5 and Figure 6). But while increasing heterogeneity leads to constant insect densities in all of the one-pathogen cases (Models 1–3), the opposite is true for the two-pathogen model (Model 4, Figure 5): as heterogeneity increases, populations move from stable populations to cycles with increasing amplitudes (Figure 5).

In all models, cycles are driven by the lag effect of increasing pathogen densities on host density over time (i.e., pathogen density continues to increase as the insect density declines, eventually causing the insect population to crash, Figure 7). However, in the host evolution model, cycles are also driven by additional evolutionary processes arising from selection for low susceptibility and from the host fecundity-resistance tradeoff. In this model, during the cycle troughs, the population is primarily made up of those low-fecundity, low-susceptibility individuals that survived the previous peak in virus density. When heterogeneity is low, the genetic composition of the host population changes more slowly, leading to large-amplitude long-period cycles as both host fecundity and pathogen transmission remain low. As the pathogen disappears, selection on transmission is weakened and the insect population booms, evolving larger average transmission because of the costs of low fecundity.

The effects of heterogeneity of transmission on insect population dynamics also depend on the extent to which this heterogeneity is determined by genetic variation in host susceptibility (Figure 6). In the host evolution model [15] (Table 1, Model 4), the heritability of transmission (which measured the fraction of variation that is explained by additive genetic effects) and insect fecundity determined whether insect–pathogen population densities remained constant or cycled over time [15]. Low to moderate heritability and high average fecundity both led to stable insect population densities. More importantly, however, by favoring resistant insects, natural selection increases insect density when the heterogeneity in transmission is high (Figure 6, bottom rows). This effect becomes stronger as more of the heterogeneity is heritable (left to right along bottom panels, Figure 6). The effect of heterogeneity on population densities will thus likely depend on the level at which this heterogeneity is determined by genetic effects.

## 4. Conclusions

Understanding the feedbacks between ecology and evolution in host–pathogen interactions is likely to be fundamental in evaluating the long-term efficacy of viruses in controlling pest insect populations. The modelling results presented here suggest that heterogeneity in transmission and life-history tradeoffs, which are partly driven by evolutionary processes, are critical factors driving insect densities. Furthermore, these processes also affected other long-term population characteristics, such as cycle amplitudes and periodicities. Based on these results, we make the following recommendations for the use of eco-evolutionary process models in insect biocontrol:**Models used to predict viral biocontrol outcomes should incorporate heterogeneity of transmission.** For single-strain non-evolutionary models, increasing heterogeneity of transmission led to higher insect population densities if it was not constrained by the mean transmission rate (i.e., there was no tradeoff; Figure 4a). Across all of the single strain models studied here, the population dynamics also became more stable as the heterogeneity increased. Although the mechanisms generating heterogeneity of transmission are likely to vary across insect–pathogen systems, the models used here [15,44,72,79] are general so that the effects of heterogeneity on population dynamics may be examined even when the sources of heterogeneity are not entirely understood.**Use wild-collected insects from local populations to test potential control agents.** Heterogeneity in transmission depends in part on host genetic variation, so experiments using a lab strain of insects is likely to misrepresent heterogeneity as well as mean transmission. In addition, the amount of variation in insect susceptibility that is explained by host genetic versus environmental effects can result in different long-term outcomes as genetic variation affects the potential for evolution of host susceptibility (and thus transmission) by natural selection. In contrast to non-evolutionary single-strain models, when evolution of host susceptibility was included, higher heterogeneity led to lower average insect densities (Figure 4b), due to the tradeoff between average transmission and insect fecundity. However, cycles are more frequent over a wider range of heterogeneity values for the evolutionary model, compared to the non-evolutionary models. Thus, it is important to consider evolutionary processes, which in turn requires considering the existing genetic variation in the targeted populations.**When possible, selection of particular virus strains for biocontrol should consider empirical measurements of heterogeneity of transmission and not just mean transmission rate.** Specifically, we recommend avoiding laboratory dose response experiments in selecting or developing a particular pathogen strain for use as a control agent. While these dose response experiments might be a good proxy for the average pathogen transmission rate, they do not contain information on heterogeneity of transmission, which surprisingly has even stronger implications for a strain’s success as a control agent (Figure 4a). Even when it is not possible to directly estimate heterogeneity of transmission using controlled epidemic experiments [44], the observed tradeoff between mean transmission and heterogeneity in transmission implies that low-mean-transmission, low-heterogeneity strains might be more effective for biocontrol than the ‘stronger’ high-mean-transmission, high-heterogeneity strains that would typically have a low lethal dose (LD50) in laboratory dose response experiments. Thus, a range of low and high LD50 strains should be field-tested before selecting a control agent, but future studies should focus on characterizing the relationship between mean transmission and heterogeneity in transmission [73].**Using multiple pathogen strains simultaneously might be better or worse than a single strain for controlling insect populations.** In our model simulations, average insect density was depressed only when the second viral strain was quite phenotypically different from the first, with a heterogeneity of transmission at least three times greater than the first strain. In other words, biocontrol using multiple pathogen strains is less effective than using a single strain if the pathogen strains have similar transmission rates and heterogeneities of transmission. This is because strains that are similar in their transmission characteristics effectively compete for the same hosts. Moreover, even in cases where two strains caused a decreased average insect population, lower mean population density was also associated with more extreme population cycles. These results thus suggest that the viral strains used in biocontrol programs should be selected based on differences in transmission characteristics, while keeping in mind the ecological risks of altering long-term characteristics of population dynamics.

### Limitations and Need for Future Studies

Although we provide a general framework to understand the effects of heterogeneity in transmission and tradeoffs on insect densities, model predictions of long-term population dynamics are sensitive to the effects of other ecological processes. For example, larger cycle amplitudes (i.e., larger insect densities) are achieved when combining the evolutionary model with environmental stochasticity and predator-insect prey dynamics [15,63]. Other approaches to model heterogeneity may provide insights into the effects of different assumptions about the heterogenetiy in transmission on long-term insect–virus population dynamics. For example, individual based models and stochastic models of disease spread [109] could drastically alter characteristics of cycling dynamics, as stochastic effects during small population sizes influence extinction events and the timing of insect outbreaks. Because the specific sources of heterogeneity in transmission can result in different long-term outcomes, pilot biocontrol studies would benefit from assessing how much of the variation in insect susceptibility is explained by environmental factors such as spatial structure, food availability and by genetic variation among insects. Quantifying the causal factors generating heterogeneity will be fundamental in inferring the role that natural selection and environmental variation play in driving the population dynamics of host–pathogen interactions and biocontrol programs. In general, evaluating the long-term efficacy of biocontrol programs should be supported by extensive ecological research to understand system-specific ecological processes.

Similarly, while our discussion focused on heterogeneity, tradeoffs, and pathogen diversity, other evolutionary processes are likely to affect the population dynamics of insect–pathogen interactions. Local adaptation has been found in some insect–virus associations [20] but not others, [79], suggesting that the use of different viral strains in bicontrol can have different outcomes across the spatial distribution of the insect pest [13]. Drift is another evolutionary process that we did not consider but that could have unexpected outcomes for biocontrol programs. Drift likely acts both within and among hosts. Among hosts, drift could be strongest after a reduction in population size following an epidemic, and even though the surviving population may be characterized by a low average transmission, further changes to the mean transmission in the population could occur through drift. For the pathogen, drift is known to act within hosts during colonization [28,39], but the role of within-host genetic drift on long-term population dynamics is unclear.

Although tradeoffs between susceptibility and fecundity have been observed in some invertebrate-pathogen systems [15,110], the evolution of host susceptibility could also occur by coevolutionary arms-races between host and pathogens. The effect of such coevolutionary processes on insect–pathogen population dynamics are however poorly understood [111,112] and further experimental and theoretical work is needed to examine the relative effects of life-history tradeoffs and arms-race evolution on the population dynamics of pest insects and their pathogens.

Studies of the evolutionary ecology of insect–pathogen interactions have shown that the success of lethal viruses in infecting pest insect hosts does not depend exclusively on the transmission rate. Variation in immune systems is ubiquitous in animal populations and insect pests are indeed characterized by their adaptability and high fitness in novel environments. Disentangling the causes of this variation by studying the evolutionary ecology of host–pathogen systems and using eco-evolutionary models as a guide to understand long-term effects is likely to help assess more precisely the efficacy of classical biocontrol.

## Figures and Tables

**Figure 4 viruses-12-00141-f004:**
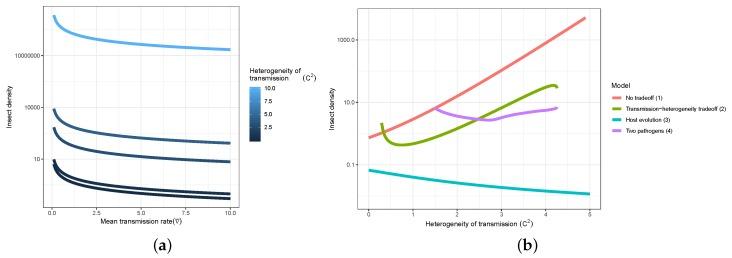
(**a**) Mean insect density ($log_{10}$) as a function of the mean transmission rate, under different values for heterogeneity in transmission $C^{2}$ (Model 1). (**b**) Effects of heterogeneity of transmission and life history tradeoffs on the long-term insect population density for four different models. Models are numbered according to Table 1; note that Model 0 (linear transmission) is not included in (**b**) as it does not include heterogeneity of transmission. The four models shown here are: “No tradeoff (Model 1)” with mean transmission held constant at $\bar{\nu }=0.7$ [72]; “Transmission-heterogeneity tradeoff (Model 2)”, with mean transmission $\bar{\nu }$ as a non-linear increasing function of heterogeneity $C^{2}$ [79]; “Host evolution (Model 3)” in which mean transmission evolves in response to a tradeoff between host susceptibility and fecundity [15]; and “Two pathogens (Model 4)”, with one pathogen strain subject to the same tradeoff as in Model 2, and the other strain with constant $\bar{\nu }=0.7$ and $C^{2}=0.576$. The two pathogen strain model is only shown for the range of heterogeneity values over which two-strain coexistence occurs. Models 2 and 4 do not cover the full extent of $C^{2}$ values because they are specifically constrained by the shape of the tradeoff curve shown in Box 2.

**Figure 5 viruses-12-00141-f005:**
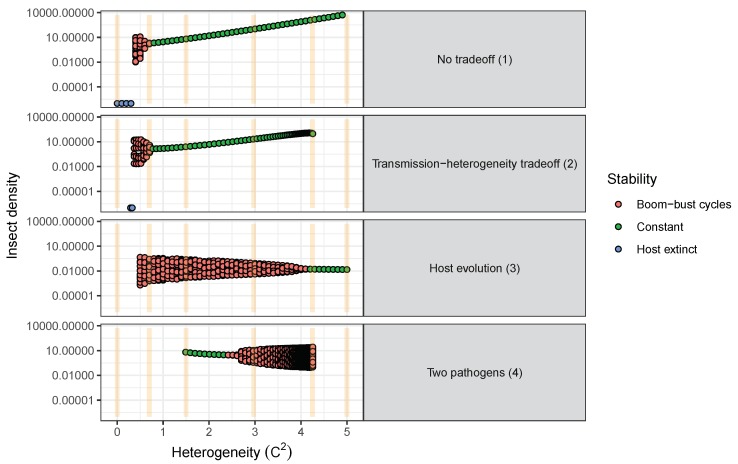
Effects of heterogeneity of transmission and life history tradeoffs on the long-term population dynamics of host insects for four different models. Models are numbered according to Table 1; note that Model 0 (linear transmission) is not included as it does not include heterogeneity of transmission. See Figure 4b for details on models and parameter values. For any given value of $C^{2}$ multiple insect densities (dots) occur whenever the insect population cycles through time. Thus, the dot range across the y-axis indicates the cycle amplitudes, whereas the number of dots indicate the frequency of recurring cycle peaks or troughs averaged over a period of 30 generations. Light orange vertical bars indicate the six heterogeneity values plotted in Figure 7.

**Figure 6 viruses-12-00141-f006:**
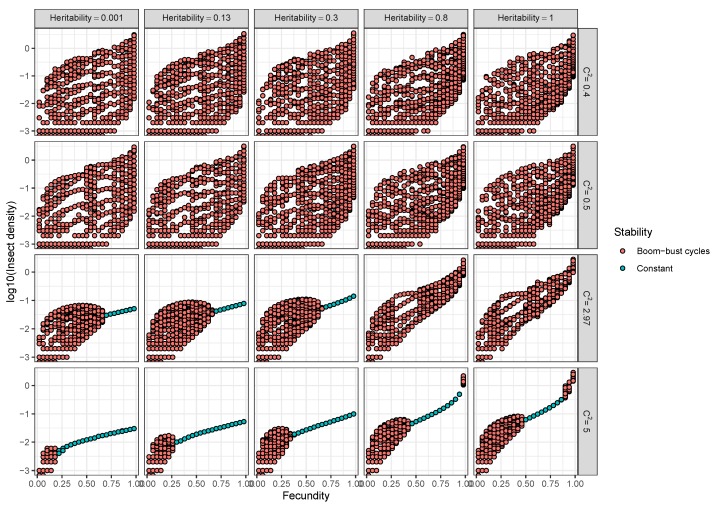
Insect density as a function of fecundity, heterogeneity of transmission, and the heritability of transmission as predicted by the evolution model (Model 3) where low transmission is costly to reproduction. For any given value of insect fecundity multiple insect densities (dots) occur whenever the insect population cycles through time. Thus, the dot range across the y-axis indicates the cycle amplitudes, whereas the number of dots indicate the frequency of recurring cycle peaks or troughs averaged over a period of 30 generations.

**Figure 7 viruses-12-00141-f007:**
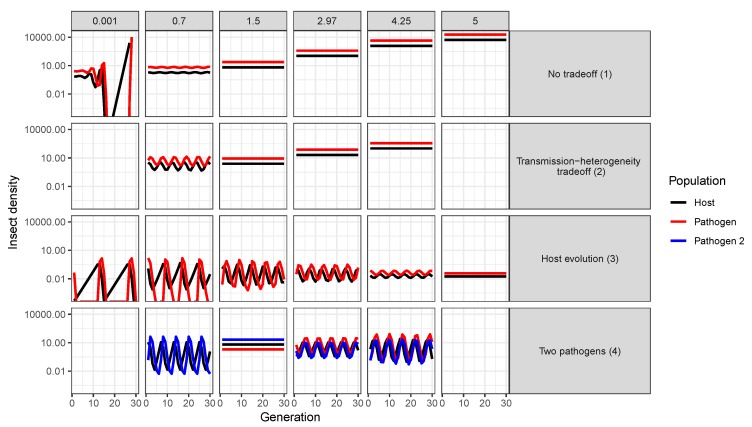
Host and pathogen population densities for four models (rows) incorporating heterogeneity of transmission $C^{2}$ (see Table 1) under realistic parameter values. See Figure 4b for details on models and parameter values. Heterogeneity $C^{2}$ is considered for five different cases (columns left to right): Very low heterogeneity ($C^{2}=0.001$), low heterogeneity ($C^{2}=0.7$), intermediate heterogeneity ($C^{2}=1.5$; the minimum value value where two-pathogen coexistence occurs in Model 4), high heterogeneity ($C^{2}=2.97$, the realistic estimate for the host evolution model (Model 3) see [15], and very high heterogeneity ($C^{2}=5$; the maximum heterogeneity considered). Note that the range of possible heterogeneity values is restricted by the transmission-heterogeneity tradeoff function for Models 2 and 4, and thus the lowest and highest cases are missing for these models. Notice also that for near-zero heterogeneity and no tradeoff (upper left corner), neither the host nor the pathogen go extinct, rather, they reach values very close to zero and so eventually the the host population booms.

**Figure 8 viruses-12-00141-f008:**
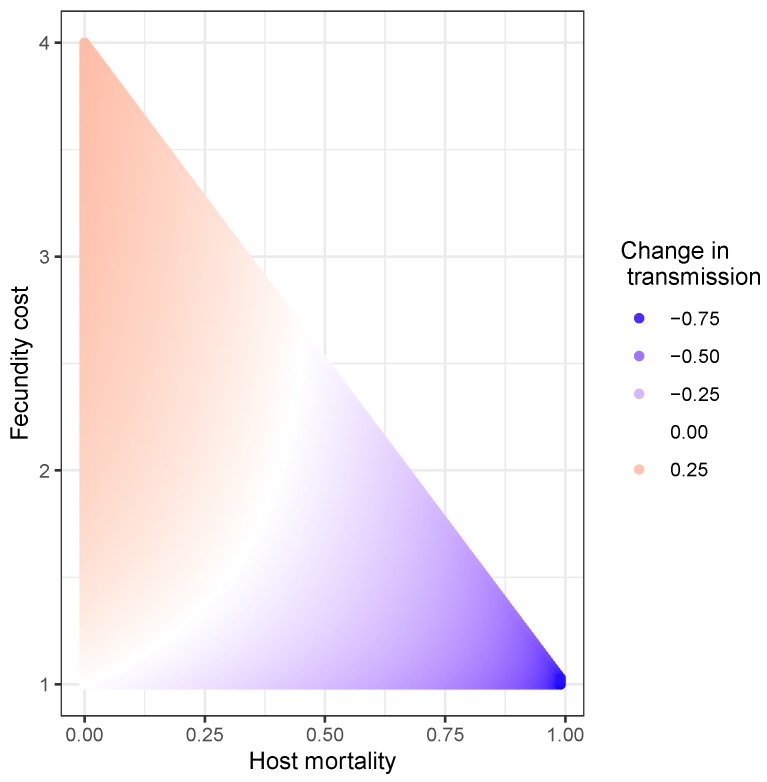
Effects of varying host survival and fecundity costs of transmission on the average transmission value of the following generation ${\bar{\nu }}_{n+1}$. Here, the change in transmission is relative to the transmission value of the previous generation, *n*, described as $\frac{{\bar{\nu }}_{n+1}-{\bar{\nu }}_{n}}{{\bar{\nu }}_{n}}$. The fecundity costs of transmission is given by the expression $1+s{\bar{\nu }}_{n}\left[1-I\left(N_{n},{\bar{\nu }}_{n},Z_{n}\right)\right]$ [15], where $I(N_{n},{\bar{\nu }}_{n},Z_{n})$ is a function describing host mortality in the previous generation, and is itself a function of the insect and pathogen population densities ($N_{n}$ and $Z_{n}$, respectively) and the transmission rate ${\bar{\nu }}_{n}$). This mortality function is proportional to the intensity of selection on transmission [15]. These results were produced by setting ${\bar{\nu }}_{n}=3$, b=0.13, $C^{2}=2.97$ and by varying *s* and $I(N_{n},{\bar{\nu }}_{n},Z_{n})$ between 0 and 1.

**Table 1 viruses-12-00141-t001:** Models used to compare effects of biocontrol on population dynamics and the equilibrium population size of host pest insects. $\bar{\nu }$ is mean transmission rate, *C* coefficient of variation of transmission, $S_{0}$ is initial density of susceptible hosts, *P* is pathogen density, *b* is heritability, *s* is the slope of fecundity and transmission, and *i* indicates pathogen strain *i*. The function $I(N_{n},{\bar{\nu }}_{n},Z_{n})$ quantifies the fraction of insects killed by virus and is thus a selective force on $\bar{\nu }$. Mode numbering begins at zero, as Model 1 converges to the linear transmission model when heterogeneity approaches zero, and thus Model (0) is not referenced later in this work.

Model Description	Force of Infection	Heterogeneity	Tradeoffs	Reference
(**0**) One pathogen strain, homogeneous host, no evolution	νPS	None (linear transmission described by ν)	None	[44]
(**1**) One pathogen strain, Gamma distribution of host susceptibility, no evolution, no tradeoffs	$\nu P{\frac{S}{S_{0}}}^{C^{2}}$	Host susceptibility, described by CV of transmission (*C*)	None	[44,72]
(**2**) One pathogen strain, same as (1) above but with evolutionary tradeoff between pathogen mean transmission and heterogeneity of transmission described by function $\nu (C^{2})$	$\nu \left(C^{2}\right)P{\frac{S}{S_{0}}}^{C^{2}}$	Host susceptibility, described by CV of transmission (*C*)	Evolutionary tradeoff between pathogen mean transmission and heterogeneity of transmission described by function $\nu (C^{2})$	[72,79]
(**3**) One pathogen strain, Gamma distribution of host susceptibility, host evolution	${\bar{\nu }}_{n}P{\frac{S}{S_{0}}}^{C^{2}}$	Host susceptibility described by the average transmission ${\bar{\nu }}_{n}$ in generation *n* and genetic and environmental variation in susceptibility $bC^{2}$	Selection on ${\bar{\nu }}_{n}$ imposed through host mortality and a tradeoff between host reproduction and susceptibility, described by $1+s{\bar{\nu }}_{n}\left[1-I\left(N_{n},{\bar{\nu }}_{n},Z_{n}\right)\right]$	[15]
(**4**) Two pathogen strains, Gamma distribution of host susceptibility, no evolution	$\nu_{i}\left(C^{2}\right)SP_{i}$	Host susceptibility and pathogen genotype, described by mean $\nu_{i}$, CV of transmission $C_{i}$, and interstrain transmission correlation ρ	Evolutionary tradeoff between pathogen mean transmission and heterogeneity of transmission described by function $\nu_{i}\left(C^{2}\right)$	[79]

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
