# Peer review of "Understanding the Evolutionary Ecology of host–pathogen Interactions Provides Insights into the Outcomes of Insect Pest Biocontrol"

_viruses, 2020, doi:10.3390/v12020141_

Round 1

Reviewer 1 Report

Review of Understanding the evolutionary ecology of host-pathogen interactions provides insights into the outcomes of insect pest biocontrol

The goal of this study is to consider biologically realistic models to consider possible outcomes of biocontrol agents – specifically, viruses applied to pest insect populations. The study compares simulations from deterministic models across a range of realistic parameter values to understand general patterns in virus transmission and host insect density under different conditions. The authors also incorporate evolution into one of their models, which has often been neglected in other studies – despite the fact that biocontrol efforts put strong selective pressure on intended hosts, and the fact that both insects and viruses can evolve very quickly.

The methods used here are nicely backed up by data and parameter values/ranges from previous work. This study fills a gap in the literature and, as the authors explain, should be useful to future considerations of viruses as biocontrol, particularly the importance of avoiding exclusive use of high-virulence virus strains, and experiments on genetically homogeneous hosts.

Detailed comments:

An ongoing (but easily fixed) issue in the manuscript is that the models are numbered inconsistently; it appears that the authors intended to number the models 0-4, with limited references to model 0, but there are many places where models seem to be referenced 1-5 instead. I’ve made note of several cases where this occurs, but the authors should check the manuscript and especially figure legends and captions to ensure that the model numbers are correct and consistent throughout.

Box 1 – the explanation of C, or V=C2, is not entirely clear here; the authors are more explicit in figure 6, stating that C2 is specifically ‘the squared coefficient of variation of transmission, and thus unitless’. I would suggest repeating or moving this text to Box 1 for clarity.

Table 1 – this is a nice way to lay out the different models that are considered, but the formatting here is a little hard to read, particularly the ‘heterogeneity’ column

Figure 4 – the notation in the model legend does not match the caption descriptions; please ensure that the numbers and model descriptions are corrected. The model shown in green is particularly confusing, as it is described as the “pathogen tradeoff” model in the caption but the "transmission-heterogeneity tradeoff" model in the legend. Please also clarify why simulations of this model do not cover the full range of the heterogeneity of transmission in the X-axis (presumably because of limitations due to the tradeoff?)

Line 237 – “boom bust” is hyphenated in a previous section, and also uses single quotation marks rather than double, i.e. ‘boom-bust’; either would be fine but the document should be consistent

Line 238 – Models 2-4 are referenced as one-strain models, but model 4 has two strains, according to table 1; assuming the authors intended to refer to models 1-3, the explanation of the shift in population stability makes sense

Lines 252-254 – The suggestion that extinction of insect populations could result from stochastic population crashes seems a bit forced; is there precedent for this in the natural history or biocontrol literature? As the authors note, the Lloyd-Smith et al. 2005 study specifically considered the possibility of pathogen extinction, not host extinction. If the authors wish to argue that this is at all likely to be possible, I would like to see further discussion of potential host extinction in the conclusions. Also see next comment –

Figure 5 -- 1) In column 1, model 1, it looks like the host population may go extinct and then be artificially re-introduced; is this the case? Please explain
Figure 5 -- 2) Misplaced parenthesis at the Páez et al reference; should come before “see”.
Figure 5 – 3) Model numbering does not match here - the caption should state models 2 and 4, I believe

Lines 317-320 – The difference in fecundity is presumably heritable as well - please clarify.

Box 2, Figure 6 – In the second sentence of the caption, “heterogeneity” is misspelled.

Line 365 – There is a missing space in the parenthetical here: “Table1” should be “Table 1”

Line 370 – see Box 2 and what?

Line 373 – “with increasing higher heterogeneity of transmission” – I think either “increasing” or “higher” would be sufficient

Line 381 – This is an interesting result! The heterogeneity term in the fixed train is also above 0.5; is it known whether that is also a necessary condition for two-strain coexistence? A slight expansion on this would be nice but is not necessary

Line 392 – I am not sure what the authors intended at the end of this line: “(Figure 7”

Line 396 – Note again that the model numbering is inconsistent

Figure 7 -- 1) In the caption, I believe the second sentence should reference Model 0
Figure 7 -- 2) Missing period at the end of the caption

Line 411 – There is either a missing reference or an extra comma inside the parentheses

Line 426 – “heterogeneity” is misspelled

Figure 8 -- 1) The x-axes on this figure are a little hard to read
Figure 8 -- 2) Please specify the model number according to Table 1 (3, I believe), for faster reference

Line 434 – I think you mean Figure 4a

Lines 450-451 -- Please reference a figure here; since the previous sentence is about the evolution model, I initially assumed you would be referencing the evolution model in Fig. 7, but that actually shows increased stability with higher heterogeneity

Section “Limitations and need for future study” – An additional sentence or two discussing the potential influence of spatial structure on transmission – perhaps in the paragraph that begins on line 496 – would make this section more thorough

Author Response

See attachement

Reviewer 2 Report

Review of Páez and Fleming-Davies, Viruses

Impressions:

The authors have produced a very nice synthesis of our understanding of how transmission heterogeneities affect long-term disease and population cycling in insect-virus systems, particularly in host systems with discrete generations and virus “overwintering” between host life-stages. This work could serve as an excellent resource for both early-career and established researchers and could help inform more applied studies on insect biocontrol with viral pathogens. In general, I have fairly minor comments that I hope will positively contribute to the paper, but it is already in great shape. This study will make a nice contribution to the journal, Viruses.

Main Comments:

In terms of structure, I was a little lost as to where the manuscript was heading. At first, I thought it would be a more targeted modeling study, but it turned out to be a really nice synthesis. I think the authors could consider a brief roadmap and more explicit goals for the study early on and not just in the abstract.

The authors focus on a few specific ODE models, and they do a nice job of explaining why these models are applicable and how they have certain advantages. However, there are model types outside of ODEs (e.g., individual-based models) that allow for host and pathogen heterogeneities in transmission, some of which could be more mechanistic, but some could still retain the agnostic emergence of “transmission” as a trait shared by the host and pathogen. I think the authors could spend a few sentences comparing model types and their pros/cons. For instance, a limitation of this work is that individual-based models might produce different expectations for long-term dynamics, because they are sometimes more sensitive to initial conditions. And individual-based models might be more appropriate if more quantitative details are known about the traits and processes of pathogens and hosts that lead to an emergent transmission rate.

I wonder if the authors should temper their confidence in the heterogeneity-mean transmission trade-off, because, to my understanding, this has only been measured in one study and in one study-system. In several places (e.g., L181, L396) the authors refer to this pattern as “common” or similar phrases that imply the pattern’s generality among systems. I think the authors could emphasize that there is sound logical reasoning to expect this to be a general pattern, but more empirical evidence is needed.

Similarly, the authors talk about their use of values of C that are “realistic” (e.g., legend of Fig 5). However, to my understanding, few studies have measured C empirically, correct? And most of these are limited to the gypsy moth system. Perhaps the authors could use this to their advantage and advocate for more empirical characterizations of transmission parameters across diverse systems.

One thing I found a bit confusing was the evolution (or lack thereof) of the C parameter in these models. If in one generation the pathogen is knocking out the most susceptible hosts, you’d expect a contraction in the phenotypic variation in transmission in the next generation, correct? And in some generations C might increase/decrease due to drift. But it’s not clear that C is evolving or if only the average transmission rate is evolving. Perhaps the authors could add a box with equations or some brief text that explains the actual process(es) of evolution in the models? Similarly, although the authors add the equation that dictates the fecundity trade-off, it is pretty unclear how this is implemented. I know that citations 15 and 75 are referenced, but I think the authors could briefly describe the methods here as well.

Minor Comments:

Figure 2: Y-axis label: wouldn’t this be overall transmission, or force of infection, because it is controlled by \mu, C, and S/S0? Figure 4A: Perhaps just use a discrete legend/color geom to match other figures with discrete C values? Figure 4 legend: I think the model numbers in parentheses need to be updated. Figure 5: could add “\n” to trans-het label to help the figure fit on the page L401: In the anecdotal case of OpNPV, the sprayed virus has been stored for a long time, and it is used across the wide geographic range of the DFTM. Understanding how the sprayed virus might interact with naturally-occurring viruses, which might be present in the environment at low initial densities, seems important. Would we need to produce different sprays for different regions? Would this be cost-effective? Basically, there’s not a lot of discussion of the economic viability of some of these suggestions.

Reviewer 3 Report

The manuscript (Viruses-678473) reviewed the influences of key ecological and evolutionary processes on success and failure of viruses as biocontrol agents of insect pests. This is a significant and worthy MS, but I recommend its major revision before it is considered for publication. The specific aspects that need to be considered are:

Overview – the authors must try to be as concise as possible, as I found the MS to be generally too wordy. Although I tried to shorten sentences at places (see attachment), I recommend that the authors do this themselves to avoid losing intended meaning. Introduction – several suggestions are made throughout this section to improve correctness and make it more readable. In line 59, the authors need to provide mechanisms in which viruses infect insects in order to provide credence to the statement that infection is not possible on diapausing insects (line 61). You need to also provide context for why you only considered infection of larvae, and not the other insect life stages. A table showing groups of viruses infecting insects (different Orders) and host stages they infect would be useful. Integrate lines 92-95 into the next section. Heterogeneity in pathogen transmission - Most of detail provided in 158-166 is also addressed in Box 1. You may want to enhance Box 1 instead, and remove it here. This makes the paragraph too wordy. Consider moving Table 1 before the section on line 177 for ease of reference. Ecological and evolutionary tradeoffs - Lines 257-261 do not provide any tradeoffs between fecundity and virus transmission. This is also highlighted in the first paragraph of Box 2. In a conventional sense, a tradeoff is established when one factor is enhanced at the expense of the other (see Snell-Rood et al. 2011 and Stevens and Stephens 2004). Lines 231-238 also describe a tradeoff in the conventional sense. However, in this section you use tradeoff in a totally different manner, you may need to provide a definition of a tradeoff.

Author Response

See Attachement
